# Thin Cell Layer Tissue Culture Technology with Emphasis on Tree Species

Vikas Sharma [1,*], Tanvi Magotra [1], Ananya Chourasia [1], Divye Mittal [1], Ujjwal Prathap Singh [1], Saksham Sharma [1], Shivika Sharma [2], Yudith García Ramírez [3], Judit Dobránszki [4,†] and Marcos Edel Martinez-Montero [5,*,†]

1   Molecular Biology & Genetic Engineering Domain, School of Bioengg and Bioscience, Lovely Professional University, Phagwara 144411, Punjab, India; tanvimagotra1806@gmail.com (T.M.); ananyachourasia238@gmail.com (A.C.); mittaldivye123@gmail.com (D.M.); us898230@gmail.com (U.P.S.); sharmasaksham1007@gmail.com (S.S.)
2   Biochemical Conversion Division, SSSNIBE, Kapurthala 144601, Punjab, India; shivikasharma25@gmail.com
3   Instituto de Biotecnología de las Plantas, Universidad Central Marta Abreu de las Villas, Santa Clara 54830, Cuba; yudithyanet@gmail.com
4   Centre for Agricultural Genomics and Biotechnology, FAFSEM, University of Debrecen, 4400 Nyíregyháza, Hungary; dobranszki@freemail.hu
5   Facultad de Ciencias Agrícolas, Universidad de Ciego de Ávila Máximo Gómez Báez, Ciego de Ávila 65200, Cuba
*   Correspondence: biotech_vikas@rediffmail.com (V.S.); cubaplantas@gmail.com (M.E.M.-M.)
†   These authors contributed equally to this work.

**Abstract:** An increased dependency on plant-based resources for food, shelter, and medicinal usage has increased their sustainable and unsustainable exploitation. To use this resource sustainably, plant tissue culture (PTC) is one important technology. Among different PTC techniques, thin cell layer (TCL) technology is a relatively simple and easily adaptable technique for in vitro cultures of plants. This technique uses small explants about 0.5–2 mm in thickness excised from different plant organs. It has been successfully used in the large-scale propagation of vegetables, legumes, and plants with medicinal benefits. TCL technology has proven to be effective in stimulating various organogenic responses when combined with various new methods such as nanotechnology or microtome-based explantation, especially in tree species. It is considered an important tool in plant biotechnology. Although the morphogenetic response per explant is usually higher in conventional explants, the appropriate use of plant growth regulators and geometric factors in TCL has the potential to make it more efficient and beneficial. This article provides an overview of the concept of TCL as applied to different plant species, particularly trees, since there are few, if any, summaries of TCL technology, especially in trees. This review will certainly revitalize this important technology so that it can be used effectively for successful mass propagation in the field of plant tissue culture.

**Keywords:** TCL; explants; morphogenesis; caulogenesis; proliferation; tree biotechnology

## 1. Introduction

Plants are multicellular organisms that belong to the plant kingdom (Plantae) and are generally autotrophic (green plants) with the exception of those which do not contain chlorophyll. Plants develop from a unicellular zygote into a multicellular organism through successive and coordinated cell divisions. However, if this process is disordered, a disorganized callus is formed [1]. The formation of various forms, structures, and functions of plant organs begins from the embryonic stage to the adult stage with successive events. Since processes such as growth and organogenesis can be altered to some extent due to various non-specific factors, it would be very difficult to study them at the biochemical and molecular levels [1,2]. Humans depend on plants for food, shelter, and, most importantly, for medicinal uses. Demand for biotic resources is increasing worldwide, and cropland is shrinking along with crop production. Considering these factors, we can apply plant tissue

culture techniques, especially thin cell layer (TCL) technology [3–5] for plant propagation in vitro as it promotes culture proliferation with enhanced productivity and reduced time, which is a key factor in plant cell and tissue differentiation.

The explants are of utmost importance for the performance of plant tissue culture. There are a number of factors related to the explants, such as genotype, tissue origin, age, size, and also the shape of the mother tissue, which are responsible for the success or failure of in vitro morphogenesis [2–4]. The idea of TCL was conceptualized by Tran Thanh Van in 1973 during his work on *Nicotiana tabacum* [6]. He proved and stated that the regeneration and reprogramming of an organ or embryo under in vitro conditions is possible if we isolate some layers of differentiated cells from the initial organ or tissue. TCL technology is based on a multicellular system using very small explants from different plant parts or organs (stems, leaves, roots, floral organs, internodes, hypocotyls, apical zones, or embryos) [4,5]. The characteristic of being "thin", i.e., an inoculum with the smallest possible number of cells, is of paramount importance for TCL technology because it facilitates the in situ localization of differentiation genes in target cells [7]. Based on the pattern of explant excision, two types of TCLs are distinguished: transverse TCLs (tTCLs) and longitudinal TCLs (lTCLs).

The difference between tTCLs and lTCLs is described in Table 1. Moreover, the schematic representation of TCLs can be seen in Figure 1.

**Table 1.** Difference between longitudinal thin cell layer (lTCL) and transverse TCL (tTCL).

| lTCL | tTCL |
|---|---|
| 1. Explant is excised longitudinally with a size of 1 mm × 0.5 or 10 mm. | 1. Explant is excised transversally with an approximate size of 0.2–0.5 mm in thickness. |
| 2. Longitudinal TCLs include only one type of tissues, i.e., a single layer (monolayer) of epidermal cells. | 2. Transverse TCL includes a very compact number of cells from different types of tissues (epidermal, cortical, perivascular, and parenchymal cells). |
| 3. Longitudinal TCLs are used to analyze mechanisms such as cell differentiation and organogenesis from the cell layer which is well defined, beyond the overcoming of barriers in the induction of morphogenesis. | 3. Transverse TCL is primarily used when the key purpose is to subdue the difficulty in obtaining the regeneration of organ or somatic embryogenesis. |

The TCL system allows the isolation of a precise cell or tissue layer that enables the in vitro induction of a specific morphogenetic program depending on factors such as genetic state, age, size and shape, pH of the medium, and synchrony with tightly controlled growth conditions such as light, media additives, temperature, plant growth regulator (PGR) concentration, etc. [4–7].

The ability of TCL to induce a specific morphogenic program depends on many factors. These include proper signal perception and transduction, the ability of internal genetic signaling pathways to respond appropriately to the given signal, the physiological state and origin of the initial tissue or organ, stress factors, and the apoptotic state or state of gene silencing cells within the tissue [8,9].

Over the past 48 years, TCL technology has been applied to various plant species, including trees and horticultural species, and successfully resulted in morphogenesis. The purpose of this review is precisely to introduce readers to the concept of TCL and its types, its advantages over conventional propagation techniques, and its applications in plant tissue cultures. Moreover, some examples of the tree species in which TCL has been successfully performed will be briefly summarized.

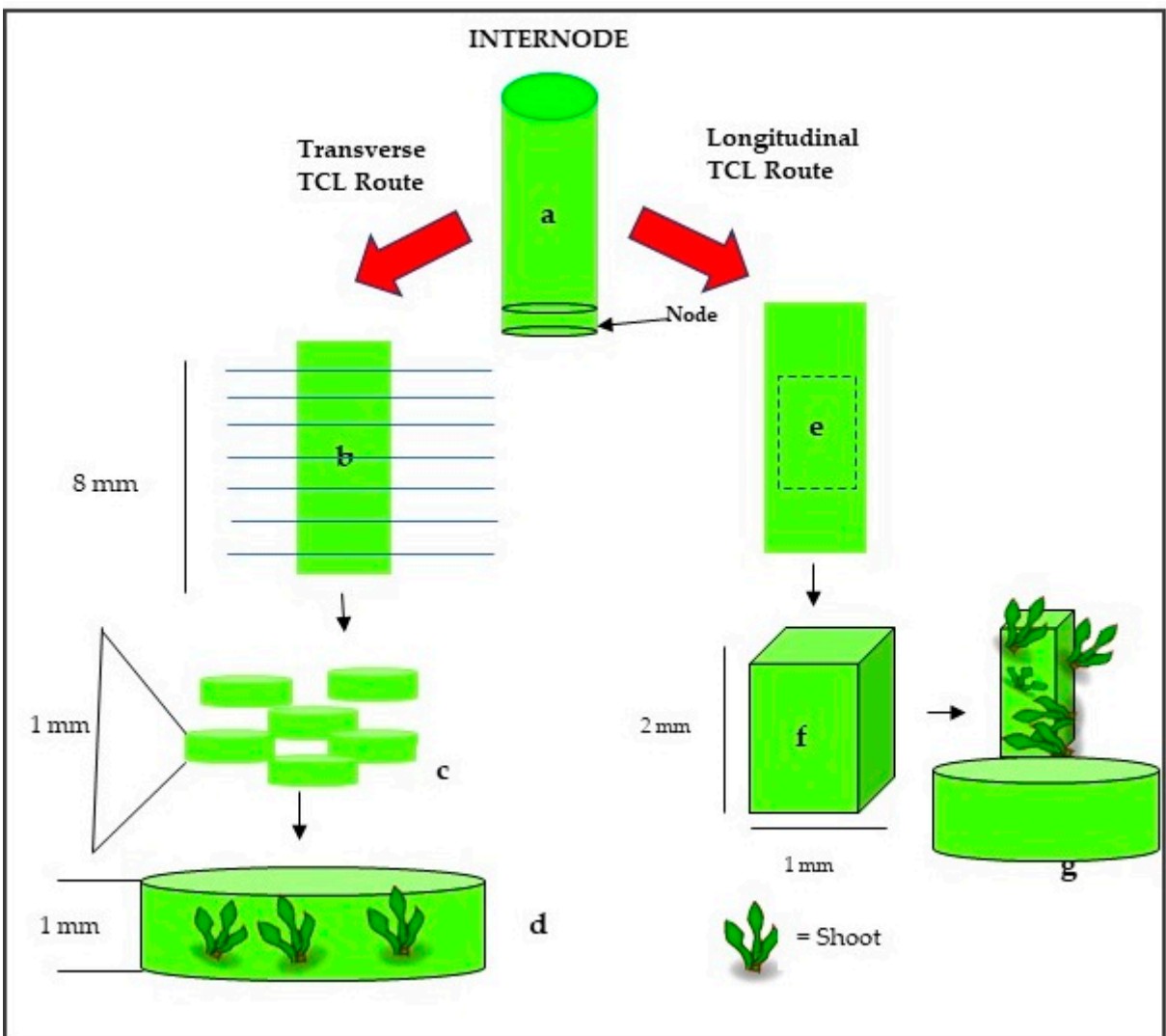

**Figure 1.** Schematic flow diagram of the preparation of a transverse thin cell layer (tTCL) or a longitudinal thin cell layer (lTCL) from a conventional explant (stem internode). (**a**) The donor explant is an internode. In the tTCL route (**b**), 1 mm thick rings are excised from the internode. (**c**) tTCLs should be cultured basal side or top side down on culture media. (**d**) Shoots will form on the tTCL explant. For the lTCL route (**e**), cut out a rectangular explant 2 mm × 1 mm in size along the internode surface. (**f**) A rectangular explant of 2 mm × 1 mm size. (**g**) Shoots are produced on the surface and not on the inner side of the internode. Modified after Teixeira da Silva and Dobránszki [4,5].

## 2. Methodology of TCL Technology

### 2.1. Explant Selection and Preparation of Thin Sections

Suitable explants (stems, nodal segments, internodes, leaves, etc.) must be selected that have better regeneration potential. Since sterilization is one of the most important steps in any plant tissue culture technique, it should also be performed carefully for TCL. Reports on sterilization of TCL are not consistent. Most investigators have suggested the preparation of thin sections from axenic mother cultures established in vitro, when surface sterilization is no longer necessary. There are reports in which material was harvested from aseptic mother stock maintained on basal medium and TCL sections were aseptically excised (*Bacopa monnieri* [10]; *Hadrolaelia grandis* [11]; *Paphiopedilum callosum* [12]; *Dendrobium* spp. [13]; *Begonia tuberous* [14]; orchid *Phalaenopsis* hybrid protocorms [15]; *Malus* spp. sprouts [16,17]). There are also reports in which general sterilization protocols with slight modifications

were used directly for thin sections of explants from field-grown plants (*Jatropha curcas* [18], and *Begonia tuberous* [14]). In these studies, the washing of explants with autoclaved double-distilled water or Milli Q water, fungicides, NaOCl, etc. has been reported. A few recent reports have also reported the use of nanoparticles to sterilize thin-layer explants and demonstrated dual efficacy [13,14].

In the previous reviews on TCL technology, very little or no emphasis has been given to the sterilization of thin-layer explants. Therefore, we have included this aspect and suggest that the seeds should be sterilized first and then the thin sections should be aseptically harvested. If seed is not available or germination is poor, explants should be from greenhouse-grown plants. Only if the first two options are not available should plants from the field be selected for thin sections. Figure 2 describes the TCL technique in detail.

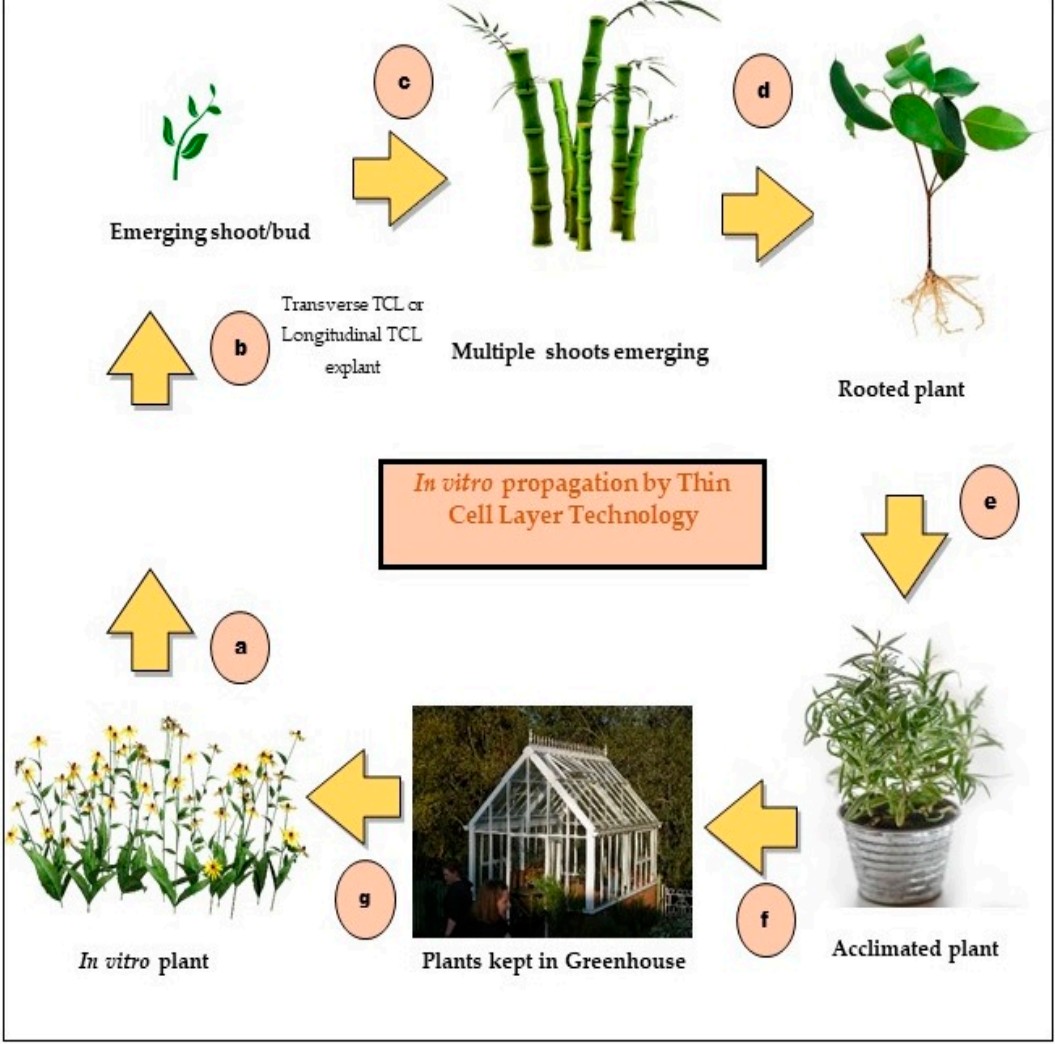

**Figure 2.** The different steps of TCL technology: (**a**) The explant of required size is excised from the mother plant and cultured on the MS medium. (**b**) A cytokinin combination is added to the MS medium to promote organ or somatic embryo development. (**c**) After several days, the plant is transferred to an elongation medium to grow several elongated shoots. (**d**) As the shoots grow, the plantlets are transferred to a root-induction medium. (**e**) The rooted plantlets are planted in a pot of garden soil for acclimatization. (**f**) The acclimatized plantlets are transferred to a greenhouse for further hardening. (**g**) In vitro plants are grown for further analysis.

## 2.2. Culture of Thin Sections

Control of the morphogenic response is the most important application of TCL. This can only be achieved by providing optimal culture conditions. The selection of an appropriate medium and its strength is therefore of paramount importance to achieve the desired morphogenic response. Aseptic thin sections should be cultured on a medium with an optimal strength and type of plant tissue culture. Most researchers have reported the use of a solid MS (Murashige and Skoog) medium [19] with full strength for the efficient cultivation of *Passiflora edulis* [20], *Gypsophyla paniculata* [13], and *Cynara scolymus* [21]. MS (Murashige and Skoog) medium (half strength) was found to be suitable for *Gerbera* spp. [22]. In Bacopa plant, the modification of MS medium with vitamins was found to be optimal for biomass generation. In addition to MS medium, woody plant medium (*Hadrolaelia grandis*) [11], Y3 (*Jatropha curcas*) [18], and modified Vacin media (*Pahiopedilum callosum*) [12] are also used in TCL. Another aspect that has been shown to be effective for TCL culture morphogenesis is the volume of the medium. In TCL from *Camellia sinensis*, the survival rate decreased as the volume of the medium increased, which could be due to the induction of hyperhydricity in a large-volume medium [23]. Based on our review, it is suggested that the use of a semi-submerged system of thin sections is optimal for better nutrient uptake and respiration than systems with liquid or solid media [12]. Although there are reports of better organogenesis in liquid cultures, we believe that the former system is comparatively easier to handle and requires fewer steps, as filter paper or a pad must be used in liquid cultures. Moreover, the incubation of cultures should be performed according to the expected morphogenic response because dark and light periods have also been reported to affect organogenesis, somatic embryogenesis (SE), and even flower production [16,17].

## 2.3. Regeneration of Roots and Shoots

The success of TCL technology also requires a more comprehensive approach to identify and intensify individual morphogenic processes. Regeneration of specific organs has been easily and effectively manipulated using TCLs, controlled specific conditions, as well as exogenously applied plant growth regulators (PGRs) [12,13]. The scaling up of the system and improved continuous quality plant production can only be achieved by careful enrichment of the selected medium with appropriate PGRs alone or in combination [5,13]. Different PGRs were used for the regeneration of roots and shoots in thin sections. For shoot regeneration, the selected media should be provided with suitable concentrations and combinations of cytokinins (kinetin, 6-benzylaminopurine (BAP), thidiazuron (TDZ), 6-($\gamma,\gamma$-dimethylallylamino)purine (2-iP), meta-topolin) and auxins (naphthaleneacetic acid (NAA), indole-3-acetic acid (IAA), indole-3-butyric acid (IBA)) [5–13]. Media with an appropriate concentration and combination of auxins such as IBA, IAA, and/or NAA can be used for root regeneration [10,12,13,18,19]. The superiority of TDZ over other cytokinins for TCL shoot organogenesis has been established in many plants (*Pahiopedilum callosum* [11], *Glypsophyla paniculata* [12], *Cynara scolymus* [21], and *Gerbera jamesonii* [22]). However, BAP was found to be effective in the organogenic development of *Bacopa* [10], *Hydrolaelia grandis* [11], *Jatropha curcas* [18], *Passiflora edulis* [20], *Ficus carica* [24], *Dendrobium* hybrids [13], and so on. An important point here is the use of additives such as coconut water, polyamines, and activated charcoal, which can also contribute to the determination of morphogenesis [11].

## 2.4. Acclimation and Hardening of Plantlets

The grown plantlets should be removed from the root induction medium without damaging the roots and washed with distilled water to remove the medium adhering around the roots. The use of distilled water is recommended because plantlets at this stage are more susceptible to microbial infection and their systemic acquired resistance is not yet fully developed to combat it [13]. After cleaning, the plantlets should be repotted into small clay pots containing substrates essential for plant growth in a natural environment,

e.g., garden soil, organic manure, sand, etc., in appropriate proportions, and kept at a temperature of 25 ± 2 °C and a humidity of 80–90% for the process of acclimation. There are reports in which different potting mixtures were used for the hardening of TCL plantlets [13,20,21,25]. In our opinion, the use of vermiculite or solarite is recommended for hardening as these materials have better water retention capacity, increased hydraulic conductivity, and it favors nutrient uptake [26].

Once the plants are acclimatized, they must be moved to a greenhouse where they can be maintained at a temperature of 28 ± 2 °C and a relative humidity of 70%–80%, or optimized according to the requirements of each plant species.

Optimization of tissue size and the application of a careful selection procedure can lead to shoot elongation and increased caulogenesis by the TCL. To this end, histological studies can be performed using the image microscope and scanning electron microscope to observe the development of pure organogenic events, meristemoids, and SE in various plants [5,13]. The genetic fidelity of TCL explants can be verified by comparing the acclimated plantlet with the mother plant using molecular marker systems such as ISSR and high-throughput techniques such as flow cytometry in *Ficus carica* [23]. The role of molecular markers in TCL plantlets has also been used to screen mutants in *Lilium longiflorum* [7].

## 3. Applications of Thin Cell Layer Technology

### 3.1. Why Is TCL Successful?

TCL technology has been successfully applied to various horticultural plants, medicinal plants, and various floral crops for in vitro propagation (Begonia, Stevia, Bacopa, tea, etc.), cryopreservation (*Calanthe davidii*), genetic transformation (Bacopa), and artificial seed production (*Cymbidium* hybrid and *Urginea altissima*). TCL technology is useful in plant biotechnology in many ways.

Figure 2 shows the different steps of TCL technology. Several reasons for the success of TCL technology are explained below.

i.　The surface area of TCL explants in contact with the media is relatively larger than that of conventional explants, which increases the efficiency of transporting the media components because they can reach more receptor or target cells. This further enhances the organogenic response compared to conventional techniques.

ii.　TCLs may be advantageous when limited plant material is available for subculture and culture establishment in in vitro cultures, as TCL explants increase the number of explants and also improve the effectiveness of subculture establishment.

iii.　Depending on the changes in the morphological and anatomical features of a plant, the preparation of a sample for light or electron microscopy by the TCL explant is much easier than the thicker conventional plants. This is because we can observe the histological condition of a plant and have the origin of an organ confirmed.

iv.　When it comes to the number of regenerating organs per initial organ, TCL technology can be more efficient than a conventional explant. Although the actual productivity of TCL explants is lower than that of conventional explant techniques, the relative productivity of TCL explants when the geometric factor (GF) and growth correction factor (GCF) were applied (as demonstrated in cymbidium hybrids, chrysanthemum, and apples) was 10- and 13-fold higher in plant species, respectively, than by conventional methods [5].

v.　The total area of stress-induced wounds in a plant is much larger in TCL explants than in conventional explants. This is also the reason that TCL explants have a higher efficiency in inducing callus formation and improving morphogenesis. When we use some organs as conventional explants, the potential of morphogenesis and organogenesis is very low. However, these conventional explants can be used to prepare TCLs for mass propagation and regeneration.

vi.　TCL technology is an excellent method for producing culturable thin sections that can be used in in vitro plant–microbe interaction assays. The use of large shoot propagule for explants in in vitro plant–microbe interaction assays is limited when used with fast

growing bacteria due to the risk of infection of growing plants in vitro. However, thin sections can overcome this limitation of in vitro bioassays developed for the screening of beneficial plant–microbe interactions [27].

### 3.2. Summary of TCL Applications on Tree Species

Research on various plant species and their results are considered applications of thin section technology. Initial reports on TCLs in tobacco have shown that the organogenesis phase can be tightly regulated in vitro in a holistic manner, which can be used to achieve complex goals such as flower development, synseed production, and the mass propagation of recalcitrant plants. We can extend this knowledge to the field of tissue culture and cultivate in a realistic and applied manner that takes into account the genetics and biochemistry of cells and organs.

Several studies have shown that protocols for plant regeneration by SE in tree species are challenging for the mass propagation of these plants [28]. Several strategies have been used to develop protocols for obtaining somatic embryos from TCLs in tree species [29,30]. Thus, there is a need to increase knowledge of SE in each tree species to increase the availability of plants for use in reforestation programs. Therefore, the use of TCL technology is proving to be an effective alternative for enhancing plant regeneration through the use of somatic embryos in angiosperms and gymnosperms (Table 2).

**Table 2.** Summary of tree species examples where TCL technology has been successfully applied.

| Plant Species | Type of TCL | Size | References |
|---|---|---|---|
| *Gymnospermae* | | | |
| *Pinus kesiya* (Royle ex. Gord) | tTCL | 1.5–2.0 mm thick | [27] |
| *Pinus patula* Scheide et Deppe | tTCL | 0.5–1.0 mm thick | [31] |
| *Pinus patula* Schl. et Cham | tTCL | 0.3–0.5 mm | [32] |
| *Pinus roxburghii* | tTCL | 0.5–1.0 mm | [28] |
| *Pinus wallichiana* | tTCL | 0.5–1.0 mm | [33] |
| *Angiospermae* | | | |
| *Citrus spp.* | tTCL | 0.4–0.5 mm | [9] |
| *Malus spp.* | tTCL | 0.1–0.3 mm | [16,17] |
| *Jatropha curcas* L. | tTCL | 0.8–1.0 mm | [18] |
| *Passiflora edulis* Sims | tTCL | 0.5 mm thick | [20] |
| *Camellia sinensis* L. | lTCL | 0.5 mm | [26] |
| *Elaeis guineensis* × *Elaeis oleifera* | tTCL | 1 mm thick | [2] |
| *Guadua chacoensis* (Rojas) Londoño and P.M. | tTCL | 0.5–1.0 mm | [34] |
| *Actinidia chinensis* | tTCL | 0.5–1.0 mm thick | [35] |
| *Bactris gasipaes* Kunth | tTCL | 0.3 mm | [36] |
| *Bactris gasipaes* Kunth | tTCL | 7.0–9.0 mm | [37] |

### 3.3. Gymnospermae

Although the use of TCLs in gymnosperms remains a challenge, TCLs were shown to develop in conifers in some Pinus species [28,29,31–33,38]. Along these lines, Malabadi et al. [28] reported in *P. kesiya* (Royle ex. Gord) that they achieved 90% calli with embryogenic structures at a thickness of 1.5–2.0 mm when they used a DCR culture medium [39] supplemented with 0.2 g/L polyvinylpyrrolidine (PVP), 7 g/L agar, 30 g/L maltose, and 0.2, 0.3, or 0.4% activated charcoal without a growth buffer. Malabadi and Van Staden [33] observed the formation of callus with embryonic structures 0.5–1.0 mm thick in Pinus patula Scheide et Deppe in a DCR-basal medium with inorganic salts at full concentration

and 0.2 g/L polyvinylpyrrolidine (PVP), 1.5 g/L gellan gum, 90 mM maltose, and 0.3% activated carbon, without growth regulators.

On the other hand, in the study by Ramirez-Mosqueda et al. [32], ITCL and tTCL segments 0.3 to 0.5 mm in thickness of immature embryos were examined in the induction of somatic embryos of *Pinus patula* Schl. et Cham in a WPM medium supplemented with 10 g/L sucrose, 1 mg/L 2,4-D, and 1 mg/L BAP. This method increased the percentage of calli with embryonic structures by 88.7% and 90% of tTCL and lTCL, respectively. Although TCL was developed for conifers, this method has not proven successful. Therefore, the search for alternatives must be deepened to address the shortcomings encountered in plant regeneration by somatic embryogenesis protocols in these species.

*3.4. Angiospermae*

In most angiosperms, TCL technology has been successfully applied to plants propagated in vitro by somatic embryogenesis. In studies by Mendoza-Pena and Hvoslef-Eide [18], an efficient protocol was developed to induce SE (somatic embryos) in adult plants of *Jatropha curcas* L. from the TCLs of young petioles to achieve massive multiplication in a plant culture medium. In their results, they found that 100% calli with embryogenic structures were achieved at a thickness of 0.8–1.0 mm. A genotype-independent protocol for callus induction and SE using tTCLs in adult *Jatropha curcas* plants was described by Mendoza-Pena and Hvoslef-Eide [18], using 2,4-D and BAP in MS and Y3 medium. A semi-strong Y3 medium enriched with 0.5 mg/L 2,4-D in combination with 0.5 mg/L BAP resulted in 100% calli induction and embryogenic responses in all accessions. Interestingly, calli started in the central part of the petiole and then spread across the explant. Another factor not previously discussed was the effect of changing the basal medium for embryo development, because when embryos were switched to a medium at half strength (MS) and 2.0 mg/L (BAP), maturation began under a 16 h/8 h photoperiod.

The superiority of the TCL technique for somatic embryogenesis was demonstrated in the hybrid *Elaeis guineensis* × *E. oleifera* [29]. An average of 30 to 50 somatic embryos were formed from a single plant (five explants), whereas three to four zygotic embryos would be required for the same number of somatic embryos. The study also revealed that 2,4-D at a concentration of 250 μM is useful for callus induction and that auxins in combination with cytokinin are essential for callus propagation with TCL technology. The reduced procedural steps resulted in less use of PGRs until the conversion of somatic embryos into plants, and reduced the cost by up to 20% [29].

Tea (*Camellia sinensis*) is one of the most industrially processed plants. Traditionally, as many other tree species, it has been propagated for commercial use. Utilization of tissue and organ societies, clonal mass propagation of elite tea, and the application of capable technologies to improve quality and yield are needed. Although many studies have been published, minimal results have been obtained in micropropagation of tea [25].

Using the TCL method, in which the first explants are shoots bearing three to four dormant buds in the late reproductive stage, a new and simple approach for tea plant propagation was successfully implemented. In this study, the volume of the medium played a crucial role, and it was found that the survival rate of TCL explants decreased as the volume of the medium increased. This is due to the fact that respiration plays an important role in the growth and development of tissues required for respiration-based protein, lipid, and carbohydrate synthesis [25].

One study demonstrated that TCL explants should be half-submerged in the media for optimal respiration and nutrient uptake. Moreover, lTCLs (0.5 mm internode length) cultured on a medium containing BA, IBA, and TDZ showed callus regeneration after four weeks [25].

Researchers also achieved improved shoot organogenesis in liquid media. Specifically, a filter-paper bridge was used with MS media in liquid at half strength containing TDZ, IBA, and BAP at different concentrations [25]. The tTCL explants were then subcultured on a solid medium after a period of six weeks, and approximately 15 shoots per tTCL explant

were obtained within 10 weeks. In addition, shoots were also formed from the calli on the semi-solid medium MS with BA and IBA.

Hanh et al. [37] evaluated the efficiency of somatic embryogenesis and regeneration from the explants of petiole and the main leaf vein of the *Actinidia chinensis* planch. using TCL technology and compared the propagation efficiency. For this purpose, they used explants of the main leaf vein (mv—1 mm × 10 mm in size) and petiole (p—1 mm × 10 mm in size) of *A. chinensis* in vitro from transverse and longitudinal TCL (mv-tTCL and mv-lTCL for mv explants, respectively, and p-tTCL and p-lTCL for p explants, respectively) with different sizes ($\frac{1}{2}$, $\frac{1}{4}$, 1/6, 1/8 of mv-lTCL and p- lTCL, respectively, and $\frac{1}{2}$, 1/3, $\frac{1}{4}$ of mv-tTCL and p-tTCL, respectively) were sectioned. Explants were placed in MS-culture medium containing 0.02 mg/L NAA, 0.5 mg/L TDZ, 30 g/L sucrose, and 8 g/L agar, and achieved the highest values in the number of somatic embryos (98.67%).

Giacomolli Polesi et al. [31] found that the TCL longitudinal technique (0.5–1 mm) to induce somatic embryos in *Guadua chacoensis* favored the percentage of callus formation with embryogenic structures (87.5%). Therefore, the results obtained by these authors serve as a basis for establishing plant regeneration protocols in other bamboo species for commercial reforestation purposes. In *Bactris gasipaes*, Campos-Boza et al. [36] reported that LCTs promoted the induction of somatic embryogenesis by 97%. Moreover, this method favored the induction of vigorous somatic embryos. Similar results were obtained by Ree and Guerra [37] in this species. The development of TCL technology in angiosperms contributes to future research related to the production of metabolites and specific genes associated with somatic embryogenesis, which would facilitate the development of efficient protocols in various tree plant species of major forestry and nutritional importance.

### 3.5. Problems and Troubleshooting

Selection of an appropriate size explant for TCL is essential. The size of the section is as critical to successful experimental results as are the area and volume. TCL thickness has been shown to be important for the shoot regeneration rate [4,5,13]. Therefore, careful attention should be paid during explantation to achieve the correct size.

Sterilization remains an important component of tissue culture experiments. The explants from thin cell sections are difficult to handle and thus need to be sterilized. Therefore, it is recommended to cut thin sections from disease-free micropropagated plants. If micropropagated plants are not available, large segments can also be surface-sterilized [13].

As mentioned earlier, thin section cutting is essential because it provides comparatively more control over the organogenic program than conventional explants, so the blade used, its sharpness, and the timing of change are among the key factors. No more than 50 explants should be prepared in one session (1 h) as longer sessions can lead to contamination and explant deterioration, and it is recommended that the blade be changed every 10–20 explants to maintain the sharpness of blades for efficient explantation. [4,13].

Immediate inoculation of thin sections is recommended because their regenerative potential and organogenic programming are compromised if they are left for half an hour or longer. For tree plants, it is recommended that tTCL explants should be oriented in an upright position to get an increased shooting frequency as this position and orientation enhances the formation of shoot meristems such as apical structures [5].

For the production of large biomasses of medically important plants, the liquid system is recommended (if hyperhydricity is not observed) as it bypasses the specific orientation of explants, thus requiring less time for hooding by the tissue culturist, cost of agar gel, and better hardening [13]. TCL technology is very efficient as it allows high productivity and less time is required for culture proliferation, which is crucial for cell differentiation tissue development [4].

### 4. Conclusions

Many studies have already been performed on TCL technology, which proves that TCL techniques are some of the most important tools that allow us to understand processes such as micropropagation, morphogenesis, and regeneration. We only need a tiny part of a particular plant and a small volume of medium to study all the elementary features responsible for regeneration and transformation, which facilitates the implementation of this technique in the laboratory under an in vitro environment. In the past, TCL methods have been developed for numerous explants and successfully applied to various plant species. The advancement of TCL technology in combination with microtome-based explantation or nanoparticle sterilization has now opened up possibilities to propagate recalcitrant plant species in vitro, especially trees that are difficult to propagate. The combination of TCL technology with bioreactor technology, histology, gene transfer, in vitro bioassays, and fidelity analysis describes the importance of this technique in plant biotechnology, especially with regard to trees. Based on the results reviewed, we conclude that this technology is distinct from other plant-based technologies and when used in combination with high-throughput technologies, it may provide excellent results in tree propagation and improvement.

**Author Contributions:** Conceptualization, V.S. and M.E.M.-M.; writing—original draft preparation, V.S., T.M., A.C., D.M., U.P.S. and S.S. (Saksham Sharma); writing—review and editing, V.S., S.S. (Shivika Sharma), Y.G.R., J.D. and M.E.M.-M.; visualization, V.S, J.D. and M.E.M.-M.; supervision, V.S.; project administration, M.E.M.-M. All authors have read and agreed to the published version of the manuscript.

**Funding:** This research received no external funding.

**Data Availability Statement:** Data are contained within this article.

**Acknowledgments:** Project no. TKP2021-EGA-20 (Biotechnology) has been implemented with support provided from the National Research, Development and Innovation Fund of Hungary, financed under the TKP2021-EGA funding scheme.

**Conflicts of Interest:** The authors declare no conflict of interest.

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
