# Peer review of "Thin Cell Layer Tissue Culture Technology with Emphasis on Tree Species"

_forests, doi:10.3390/f14061212_

Round 1

Reviewer 1 Report

Sharma et al. presented a review on the employment of Thin Cell Layer (TCL) Technology with a special emphasis on tree species. As there are limited reviews on the topic, it can be beneficial for the readers. The manuscript is nice, however, there are few points that should be considered to increase the value of the manuscript and may be readability.

-A major lack in the manuscript is that the first paragraph of the manuscript does not direct towards the need of this technique. Authors must add a separate heading mentioning why TCL is required in today’s scenario. If plants can be simply grown by normal method, why such technology is required.

-Similarly, the need of this review should be highlighted in the abstract.

-Figure 2, please add the full form of tTCL and lTCL in the caption.

-In conclusion, the authors mention ‘The combination of TCL with bioreactor technology, histology, gene transfer, in vitro bioassays, and fidelity analysis describes the importance of this technique in plant biotechnology.’ Separate headings should be added to discuss the employment of TCL with each one of these techniques.

Moderate editing of English language is required. 

Author Response

Dear Reviewer 1,

First, we would like to thank you for your work on our manuscript and your helpful suggestions. We have made some changes in the text as a result. For your convenience, all changes have been highlighted in the manuscript using the Word editing option MS so that you can easily find all corrections according to your and another reviewer's comments.

Point 1: -A major lack in the manuscript is that the first paragraph of the manuscript does not direct towards the need of this technique. Authors must add a separate heading mentioning why TCL is required in today’s scenario. If plants can be simply grown by normal method, why such technology is required.

Response 1: Thanks for your valuable suggestions. We have added “Considering these factors, we can apply plant tissue culture techniques, especially thin cell layer (TCL) technology [3-5] for plant propagation in vitro, as it promotes culture propagation with increased productivity in shorter time, which is a key factor in plant cell and tissue differentiation." In addition, we have already added the section ¨Why TCL is successful¨. This summarises the advantages of TCL over conventional methods and outlines its requirements.

Point 2: -Similarly, the need of this review should be highlighted in the abstract.

Response 2: The abstract has been revised.

Point 3: -Figure 2, please add the full form of tTCL and lTCL in the caption.

Response 3: The full form has been added to the caption.

Point 4: -In conclusion, the authors mention ‘The combination of TCL with bioreactor technology, histology, gene transfer, in vitro bioassays, and fidelity analysis describes the importance of this technique in plant biotechnology.’ Separate headings should be added to discuss the employment of TCL with each one of these techniques.

Response 4: Dear Reviewer thank you very much for the valuable suggestion. The authors have already included the recent studies and references in the paper. Also, we have already included the references in which nanotechnology was used for sterilisation of explants. In addition, we have included the problem of sterilisation in the culture of TCLs. Since this review refers to tree species, we could not find much data on such work using non-technologies affecting TCL. Biorector technology with TCL cultivation has been used in some non-tree species such as Bacopa, which we believe is outside the scope of the current special issue, and we are in the process of addressing this topic separately in another article. The in vitro assay for plant microbes has been established by our group, and we are conducting experiments with TCLs that are initially promising, but to draw conclusions we need more concrete studies, which are lacking at the moment. Therefore, in this article, we considered these techniques in a futuristic way where TCL could be effective.

Thank you again for your help and suggestions, which have undoubtedly improved the readability and clarity of our text.

Yours sincerely,

Authors

Reviewer 2 Report

Thin cell layers technology has been used in plant tissue culture for more than 48 years. This article introduces the reader to the concepts and types of TCL, its advantages over traditional propagation techniques, and its application in plant tissue culture. As far as I know, the previous review on thin cell layers was published in 2015. Seven years have passed. We hope to see something new. For example, TCL technology combined with nanoparticle sterilization technology, TCL and bioreactor technology, TCL how to apply in vitro plant - microbial interaction test; In addition, the problems in widespread use of TCL should also be summarized.

Line 109-111 The author emphasizes that compared with previous reviews, the sterilization of thin-layer explants is added, but there is not much introduction. The sterilization method of nanoparticles mentioned by the author is not introduced in detail.

Line 132-134 “we believe that the former system is comparatively easier to handle and requires fewer steps, as filter paper or a pad must be used in liquid cultures ", is there a good case to prove it?

Line 169-170  “In our opinion, the use of vermiculite or solarite is recommended for hardening as these materials have better water retention capacity. ”  It is suggested to give some convincing cases

Line212-216  “The actual productivity of TCL explants is lower than that of conventional explant techniques? ” Should be discussed as a shortcoming of TCL technology.

“the relative productivity of TCL explants when the geometric factor (GF) and growth correction factor (GCF) were applied (as demonstrated in cymbidium hybrids, chrysanthemum, and apple) was 10- and 13-fold higher ", Where is this data coming from?

Author Response

Dear Reviewer 2,

First, we would like to thank you for your work on our manuscript and your helpful suggestions. We have made some changes in the text as a result. For your convenience, all changes have been highlighted in the manuscript using the Word editing option MS so that you can easily find all corrections according to your and another reviewer's comments.

Point 1: Thin cell layers technology has been used in plant tissue culture for more than 48 years. This article introduces the reader to the concepts and types of TCL, its advantages over traditional propagation techniques, and its application in plant tissue culture. As far as I know, the previous review on thin cell layers was published in 2015. Seven years have passed. We hope to see something new. For example, TCL technology combined with nanoparticle sterilization technology, TCL and bioreactor technology, TCL how to apply in vitro plant - microbial interaction test; In addition, the problems in widespread use of TCL should also be summarized.

Response 1: Dear Reviewer thank you very much for the valuable suggestion. The authors have already included the recent studies and references in the paper. Also, we have already included the references in which nanotechnology was used for sterilisation of explants. In addition, we have included the problem of sterilisation in the culture of TCLs. Since this review refers to tree species, we could not find much data on such work using non-technologies affecting TCL. Bioreactor technology with TCL cultivation has been used in some non-tree species such as Bacopa, which we believe is outside the scope of the current special issue, and we are in the process of addressing this topic separately in another article. The in vitro assay for plant microbes has been established by our group, and we are conducting experiments with TCLs that are initially promising, but to draw conclusions we need more concrete studies, which are lacking at the moment. Therefore, in this article, we considered these techniques in a futuristic way where TCL could be effective.

Point 2: Line 109-111 The author emphasizes that compared with previous reviews, the sterilization of thin-layer explants is added, but there is not much introduction. The sterilization method of nanoparticles mentioned by the author is not introduced in detail.

Response 2: At this point we would like to explain that in previous studies little or very little information was given by the authors, just as in a normal article on plant tissue culture. For this reason, we have tried to give some tips and tricks for explantation and sterilisation. The sterilisation method with nanoparticles is the same as other sterilisation agents, as the plants are cut afterwards. Therefore, we did not discuss much about this content, but focused on the TCL.

 Point 3: Line 132-134 “we believe that the former system is comparatively easier to handle and requires fewer steps, as filter paper or a pad must be used in liquid cultures ", is there a good case to prove it?

Response 3: Dear Reviewer, thank you very much for these valuable questions. There are no direct studies comparing submerged and liquid culture systems in conjunction with TCL. These suggestions are based on our observations in handling TCL cultures in these two types of systems.

Point 4: Line 169-170  “In our opinion, the use of vermiculite or solarite is recommended for hardening as these materials have better water retention capacity. ”  It is suggested to give some convincing cases

Response 4: The reference has been added.

Point 5: Line212-216  “The actual productivity of TCL explants is lower than that of conventional explant techniques? ” Should be discussed as a shortcoming of TCL technology.

Response 5: Thank you again for your valuable comments, but in this section we give a comparative report with relative productivity higher than conventional methods as reported for the mentioned plant species.

Point 6: “the relative productivity of TCL explants when the geometric factor (GF) and growth correction factor (GCF) were applied (as demonstrated in cymbidium hybrids, chrysanthemum, and apple) was 10- and 13-fold higher ", Where is this data coming from?

Response 6: The reference for the above part has already been given.

Thank you again for your help and suggestions, which have undoubtedly improved the readability and clarity of our text.

Yours sincerely,

Authors

Reviewer 3 Report

- The citations are not sufficient for a review article. Furthermore, new citations (2023 references) are needed.

- The title of the manuscript has emphasis on tree species. But there is no information and content about trees in some parts of the manuscript specially in the conclusion.

- The scientific names of plant species must be in italic in the bibliographic list.

Author Response

Dear Reviewer 3,

First, we would like to thank you for your work on our manuscript and your helpful suggestions. We have made some changes in the text as a result. For your convenience, all changes have been highlighted in the manuscript using the Word editing option MS so that you can easily find all corrections according to your and another reviewer's comments.

Point 1: - The citations are not sufficient for a review article. Furthermore, new citations (2023 references) are needed.

Response 1: We have added the most relevant and latest references up to 2022.

Point 2: - The title of the manuscript has emphasis on tree species. But there is no information and content about trees in some parts of the manuscript specially in the conclusion.

Response 2: Dear reviewer, thank you for your valuable comment. We have focused on the TCL studies on trees and have included the data in the form of text and tables. Only in the parts where we need to discuss TCL as techniques or problems, we have not included them.

Point 3: - The scientific names of plant species must be in italic in the bibliographic list.

Response 3: Done

Thank you again for your help and suggestions, which have undoubtedly improved the readability and clarity of our text.

Yours sincerely,

Authors

Reviewer 4 Report

Dear authors,

Please, go through the manuscript and make amendments to the phrase, comments, and suggestions below.

Line 16 – 17: “vegetable legumes and plants”. Make this sentence clear that if the sentence would have been vegetable, legumes, and plants?

Line 30 – 31: This sentence needs to be paraphrased “Plants are multicellular organisms that belong to the plant kingdom (Plantae) and are generally autotrophic.’ Not necessarily all plants are autotrophic.

Line 43 – 44: Omit the first conditions, as the second on line 44 represents both.

Line 47 – 48: The following sentence needs to be cited.  The idea of TCL was conceptualised by Tran Thanh Van in 1973 during his work on Nicotiana tabacum.”

Line 45: smallest possible number or layer?

Line 79: Spell out PGR.

Line 91: will be briefly sumarise (sumarised).

Line 120 and 121: Spell out MS?

Line 151: “The superiority of TDZ over other cyto-…” It is the ratio of auxin /cytokinin, not superiority (that apparently shows the quality of the hormones). Previous research works indicated that work has shown that a high auxin/cytokinin ratio induces root regeneration, whereas a low ratio promotes shoot initiation. Paraphrase this sentence and research articles that founded the above statements.

Line 218: “The total area of stress-induceyd . . .” stress-induced ~ spelling error?

Line 230: I think the abbreviation of TLC (underlined in red in the manuscript) stands for thin cell layer (TCL). Check all that apply after line 230.

Line 238: What is SE stands for?

Line 258 and 262: What is DCR stand for?

Line 353 – 354: session (1 h) and changed . . . Please, add justification:

1. time of the session is to be limited to 1 h.

2. the blade changed every 10 -20 explants?

Line 360: medically or medicinally? I think medicinally is the appropriate word for this specific context.

With regards,

The reviewer

General remarks:

·        Abbreviated acronyms must be spelled out at their first appearance.

·        Some abbreviations are either not spelled or misspelled, and you need to correct the red underlined throughout the manuscript.

·        Some punctuation needs to be appropriately used.

·        Choose the best contextual words for a better understanding of the contents of the manuscript.

Author Response

Dear Reviewer 4,

First, we would like to thank you for your work on our manuscript and your helpful suggestions. We have made some changes in the text as a result. For your convenience, all changes have been highlighted in the manuscript using the Word editing option MS so that you can easily find all corrections according to your and another reviewer's comments.

Point 1: Line 16 – 17: “vegetable legumes and plants”. Make this sentence clear that if the sentence would have been vegetable, legumes, and plants?

Response: Corrected

Point 2: Line 30 – 31: This sentence needs to be paraphrased “Plants are multicellular organisms that belong to the plant kingdom (Plantae) and are generally autotrophic.’ Not necessarily all plants are autotrophic.

Response: Updated

Point 3: Line 43 – 44: Omit the first conditions, as the second on line 44 represents both.

Response: Done

Point 4: Line 47 – 48: The following sentence needs to be cited.  “The idea of TCL was conceptualised by Tran Thanh Van in 1973 during his work on Nicotiana tabacum.”

Response: Reference added

Point 5: Line 45: smallest possible number or layer?

Response: Number as per the refernce taken

Point 6: Line 79: Spell out PGR.

Response: Done

Point 7: Line 91: will be briefly sumarise (sumarised)

Response: Changed.

Point 8: Line 120 and 121: Spell out MS?

Response: Done

Point 9: Line 151: “The superiority of TDZ over other cyto-…” It is the ratio of auxin /cytokinin, not superiority (that apparently shows the quality of the hormones). Previous research works indicated that work has shown that a high auxin/cytokinin ratio induces root regeneration, whereas a low ratio promotes shoot initiation. Paraphrase this sentence and research articles that founded the above statements.

Response: In our opinion, this has been observed in many studies where auxins and cytokinins were used alone or in combinations and different morphogenic responses were reported. TDZ has shown better response in some plant species alone or in combination with the same or different auxin and/or cytokinin than other cytokinins and similar combinations. In this case, we can say that one cytokinin is superior compared to others in a given series of experiments. The authors therefore believe that this should remain the same in the manuscript.

Point 10: Line 218: “The total area of stress-induceyd . . .” stress-induced ~ spelling error?

Response: Corrected

Point 11: Line 230: I think the abbreviation of TLC (underlined in red in the manuscript) stands for thin cell layer (TCL). Check all that apply after line 230.

Response: Corrected

Point 12: Line 238: What is SE stands for?

Response: Somatic embryos

Point 13: Line 258 and 262: What is DCR stand for?

Response: Thanks for your valuable comment. The DCR was formulated and used as a modified medium for the Douglous fir at that time, a full form for it wasn't given in this research manuscript. We also couldn't find it on the websites of some media providers.

Point 14: Line 353 – 354: session (1 h) and changed . . . Please, add justification:

  1. time of the session is to be limited to 1 h. 2. the blade changed every 10 -20 explants?

Response: Done

Point 15: Line 360: medically or medicinally? I think medicinally is the appropriate word for this specific context.

Response: Done

Thank you again for your help and suggestions, which have undoubtedly improved the readability and clarity of our text.

Yours sincerely,

Authors

Round 2

Reviewer 2 Report

The author gave better answers to the questions raised, I agreed to publish the article.